# Functional Characterization of Largemouth Bass (*Micropterus salmoides*) Soluble FcγR Homolog in Response to Bacterial Infection

**DOI:** 10.3390/ijms232213788

**Published:** 2022-11-09

**Authors:** Jing Wu, Yanping Ma, Yifan Nie, Jingya Wang, Guoqing Feng, Le Hao, Wen Huang, Yugu Li, Zhenxing Liu

**Affiliations:** 1Institute of Animal Health, Guangdong Academy of Agricultural Sciences, Guangzhou 510640, China; 2College of Veterinary Medicine, South China Agricultural University, Guangzhou 510642, China; 3Key Laboratory of Livestock Disease Prevention of Guangdong Province, Guangzhou 510640, China; 4Scientific Observation and Experiment Station of Veterinary Drugs and Diagnostic Techniques of Guangdong Province, Ministry of Agriculture and Rural Affairs, Guangzhou 510640, China; 5Collaborative Innovation Center of Guangdong Academy of Agricultural Sciences, Guangzhou 510640, China; 6Guangdong Key Laboratory of Animal Breeding and Nutrition, Institute of Animal Science, Guangdong Academy of Agricultural Sciences, Guangzhou 510640, China

**Keywords:** largemouth bass, soluble FcγRs, IgM-binding, ADCP, *Nocardia seriolae*

## Abstract

Fc receptors (FcRs) are key players in antibody-dependent cellular phagocytosis (ADCP) with their specific recognition of the Fc portion of an immunoglobulin. Despite reports of FcγR-mediated phagocytosis in mammals, little is known about the effects of soluble FcγRs on the immune response. In this study, FcγRIα was cloned from the largemouth bass (*Micropterus salmoides*) (MsFcγRIα). Without a transmembrane segment or a cytoplasmic tail, MsFcγRIα was identified as a soluble form protein and widely distributed in the spleen, head kidney, and intestine. The native MsFcγRIα was detected in the serum of *Nocardia seriolae*-infected largemouth bass and the supernatants of transfected HEK293 cells. Additionally, it was verified that the transfected cells’ surface secreted MsFcRIα could bind to largemouth bass IgM. Moreover, the expression changes of *MsFcγRIα*, *Syk*, and *Lyn* indicated that *MsFcγRIα* was engaged in the acute phase response to bacteria, and the FcγR-mediated phagocytosis pathway was activated by *Nocardia seriolae* stimulation. Furthermore, recombinant MsFcγRIα could enhance both reactive oxygen species (ROS) and phagocytosis to *Nocardia seriolae* of leukocytes, presumably through the interaction of MsFcγRIα with a complement receptor. In conclusion, these findings provided a better understanding of the function of soluble FcγRs in the immune response and further shed light on the mechanism of phagocytosis in teleosts.

## 1. Introduction

Fc receptors (FcRs), specific for the constant fragment crystallizable (Fc) region of immunoglobulin (Ig), are widely expressed in leukocytes, such as monocytes, macrophages, neutrophils, NK cells, and B lymphocytes [1]. In general, classical mammalian FcRs are composed of extracellular C2 Ig domains, a transmembrane (TM) segment, and a cytoplasmic tail (CYT) [2]. Besides the activation or inhibition of cellular responses, the FcRs mediate other diverse effector functions to clear immune complexes [3], particularly in antibody-dependent cell-mediated cytotoxicity (ADCC) and antibody-dependent cellular phagocytosis (ADCP) [4,5]. These processes are initiated by the cross-linking of FcRs with Ig [6]. While FcRs recognize Fc fragments, Src family kinases trigger the phosphorylation of spleen tyrosine kinase (Syk) and immunoreceptor tyrosine-based activation motif (ITAM) within CYT, then activate the phagocytosis-related genes [7,8,9].

FcRs for IgG (FcγR), IgE (FcεR), IgA (FcαR) and IgM (Fcα/µR, FcµR, pIgR) have been found in mammals [10,11]. Differently from mammals, Igs in teleost only four types (IgM, IgD, IgZ/IgT, IgM-IgZ chimera) [12], of which IgM was discovered firstly [13]. To date, polymeric immunoglobulin receptor (pIgR), one of the FcRs, in grass carp (*Ctenopharyngodon idellus*) [14], flounder (*Paralichthys olivaceus*) [15], Nile tilapia (*Oreochromis niloticus*) [16], ballan wrasse (*Labrus bergylta*) [17] and largemouth bass (*Micropterus salmoides*) [18] have been reported that could bind to IgM. However, regarding classical IgM-binding FcRs homologs, it has only been identified in FcγRI of ayu (*Plecoglossus altivelis*) [19] and FcRI of channel catfish (*Ictalurus punctatus*) [2]. The in-depth understanding of the function of FcRs in teleost is at a very early stage.

Soluble forms of FcRs (sFcRs) are generated either by proteolytic cleavage of the membrane FcRs or alternative splicing of transmembrane (TM)-encoding exon [20]. Distinguishing from classical FcRs, the structure of sFcRs homologs without TM or CYT has been confirmed in the teleosts, including ayu [19] and channel catfish [2]. Mammalian sFcRs are reported to function in the inhibition of immune complexes binding [21], down-regulation of B cell proliferation and antibody production [20], as well as interaction with complement receptors, to trigger cellular activation [22]. However, functional studies of sFcRs in teleost are currently limited to inhibiting IgM secretion [19]. Whether the sFcRs can regulate ADCP like classical FcRs is not yet well understood.

The largemouth bass (*Micropterus salmoides*) is an economically freshwater fish with commercial value [23]. In recent years, serious losses have occurred in largemouth bass aquaculture due to the emerging infectious diseases caused by *Nocardia seriolae* (*N. seriolae*) [24,25,26]. Innate immunity is the first line of defense against pathogens [27], and the role of FcRs in it deserves attention. However, little information is available about the participation in the immune response of sFcRs.

In this study, the largemouth bass *FcγRIα* (*MsFcγRIα*) gene was cloned, and then the sequence and structural characteristics were analyzed. The recombinant MsFcγRIα ((r)MsFcγRIα) protein was expressed in *Escherichia coli* and HEK293 cells. Moreover, a polyclonal antibody against MsFcγRIα was prepared to identify the localization in HEK293 cells, the native expression in *N. seriolae*-infected largemouth bass serum, and the distribution in largemouth bass tissues. The mRNA relative expression of *MsFcγRIα* and its downstream genes (*Syk* and *Lyn*) were investigated in vitro and in vivo upon *N. seriolae* and lipopolysaccharide (LPS) infection. In addition, the binding of transfected-MsFcγRIα to purified IgM was examined. Concurrently, the regulation of (r)MsFcγRIα on respiratory burst, phagocytosis, and transcriptional level of phagocytosis-related and complement system genes were detected. This research will broaden our understanding of the sFcRs functions in teleost.

## 2. Results

### 2.1. Cloning and Sequence Analysis of MsFcγRIα

The ORF of *MsFcγRIα* consisting of 1272 bp (GenBank accession No. OK258092) was successfully cloned from the head kidney cDNA of largemouth bass (Appendix A). It encoded a polypeptide of 423 amino acids (aa) with a putative molecular mass of 47.1 kDa (Figure 1). The theoretical isoelectric point (PI) of MsFcγRIα protein was 6.32. One potential N-glycosylation site (N^57^SS) was observed in MsFcγRIα. The secondary structure of MsFcγRIα was composed of a signal peptide (resides: aa 1–19) at N-terminate and two Ig domains: Ig domain 1 (D1) consisting of 83 aa (aa 35–117) and Ig domain 2 (D2) consisting 84 aa (aa 125–208) (Figure 2A). D1 contained conserved cysteine residue (Cys) forming a disulfide bond (Cys^50^-Cys^93^) (Figure 2B) and was classified into the C2 set. Moreover, there was neither a transmembrane domain nor a GPI-anchorage site, suggesting that MsFcγRIα existed as a soluble form. Similar to mammalian FcR, the predicted *β*-strand were found in MsFcγRIα D1 and D2 (Figure 2B), and the homology modeling showed a similar tertiary protein structure of FcγRIα between largemouth bass and human (PDB Hit: 3RJD) (Figure 2C).

Sequence comparison analysis showed that MsFcγRIα was tightly grouped with Asian seabass (*Lates calcarifer*) FcRIL (54.58% identity) and highly identical to large yellow croaker (*Pseudosciaena croce*) FcRLα (49.60% identity). Phylogenetic tree analysis showed that MsFcγRIα clustered independently into one branch in fish (Figure 3A). Furthermore, MsFcγRIα D1 and D2 also distinctly clustered together within counterpart groups (Figure 3B). The phylogenetic analysis indicated that largemouth bass had a closer evolutionary relationship to other FcRs sequences of fish.

### 2.2. Antibody Specificity and Subcellular Localization of MsFcγRIα

The results of SDS-PAGE showed that MsFcγRIα-PET32a was successfully expressed in *E. coli* and highly purified with a theoretical molecular mass of 64.6 kDa (Figure 4A, lines 3, 4). A mouse anti-MsFcγRIα polyclonal antibody (PcAb) was obtained.The specificity was validated by Western blotting (Figure 4A, line 5). To determine the expression of native MsFcγRIα could be recognized by the anti-MsFcγRIα PcAb, MsFcγRIα-pcDNA3.1(+) plasmid was transfected into HEK293 cells. Results showed that after 24 h transfection, 42.23 ± 0.622% of the MsFcγRIα was successfully expressed on HEK293 cells (Figure 4C,D). Moreover, a specific reaction band on the PVDF membrane was observed in *N. seriolae*-infected largemouth bass serum, supernatants, and cell lysates of transfected HEK293, respectively. (Figure 4A, lines 6, 8, 9), but no band in healthy largemouth bass serum (Figure 4A, lines 7). In short, the prepared anti-MsFcγRIα PcAb could specifically recognize MsFcγRIα.

To verify the predicted subcellular localization of MsFcγRIα, the MsFcγRIα-EGFP plasmid was transiently transfected into HEK293 cells. After 24 h, the green signal (EGFP) was localized on the cell membrane and cytoplasm (Figure 4B). All of the results suggested that the anti-MsFcγRIα PcAb could specifically recognize MsFcγRIα, and MsFcγRIα was expressed both on the cell membrane and cytoplasm.

### 2.3. Secretion of MsFcγRIα in Immune Tissues

IHC was performed to detect the expression of secretory MsFcγRIα protein in the immune tissues of largemouth bass. The PBS solution was employed in the negative control to replace anti-MsFcγRIα PcAb. IHC results (Figure 5) showed that the strong positive signals (+++) of MsFcγRIα existed in the intestine, spleen, and head kidney. Moderately positive signals (++) were mainly focused on intestinal columnar epithelial cells and goblet cells. Weakly positive signals (+) were also found among the extracellular matrix of the spleen, head kidney, and intestinal villus. It is suggested that MsFcγRIα was a secretory protein widely distributed in the immune tissues of largemouth bass.

### 2.4. Binding of MsIgM to MsFcγRIα

The binding of expressed MsFcγRIα on transfected HEK293 cells to purified MsIgM was confirmed by flow cytometric, indirect immunofluorescence, and co-immunoprecipitation (CO-IP) assay. Results of flow cytometric assay indicated that after 0 h, 6 h, 12 h, and 24 h incubation with MsIgM, the MsFcγRIα-expressed HEK293 cells combined with 3.54 ± 0.22%, 19.70 ± 0.90%, 26.33 ± 0.61% and 40.27 ± 0.24% MsIgM, respectively (Figure 6(A3)). Meanwhile, fluorescence microscope observation (Figure 6B) demonstrated that MsFcγRIα-EGFP (green signal) was uniformly distributed in the cytoplasm and membrane. Double-immunofluorescence straining showed that MsIgM could be co-located with the MsFcγRIα-EGFP expressed on the membrane. In contrast, there was no red signal of MsIgM on pEGFP-N1-transfected HEK293 cells.

To exclude the effect of EGFP protein on the binding of MsIgM to MsFcγRIα, recombinant MsFcγRIα-pcDNA3.1(+) was constructed and transfected into HEK293 cells. Following incubation with MsIgM, the interactional protein in cell lysates was precipitated by anti-MsIgM mAb (MM06H). Western blotting was detected by MM06H and anti-MsFcγRIα PcAb, respectively. Results (Figure 6C) showed that a 47 kDa protein band was found in HEK 293 cell lysates following MsFcγRIα-transfected, MsFcγRIα-transfected with MsIgM incubation (Input) as well as IP precipitate of MsFcγRIα-transfected cells with MsIgM incubation (CO-IP) detected by anti-MsFcγRIα PcAb, but no band was observed in IP precipitate of the MsFcγRIα-transfected cells without MsIgM incubation. 72 kDa identical to the molecular mass of MsIgM can be observed in the lysates of MsFcγRIα-transfected HEK 293 cells following MsIgM incubation (Input) and its IP precipitate (IP) determined by MM06H, and faintly observed in the lysate of pcDNA3.1(+)-transfected cells with MsIgM incubation (IP) and its IP precipitate (IP). Still, no corresponding band was observed in cells without MsIgM incubation.

The results indicated that the purified MsIgM could interact with the surface-expressed MsFcγRIα on transfected HEK293 cells.

### 2.5. Gene Expression Profiles in Tissues

The mRNA expression profile of *MsFcγRIα* in healthy largemouth bass tissues was explored by qRT-PCR. The results (Figure 7A) show that *MsFcγRIα* was widely expressed in all examined tissues and highly expressed in immune-related tissues, including the spleen, head kidney, and intestine. The highest abundance was detected in the spleen. Inversely, the expression of *MsFcγRIα* in the brain, liver, and gill was lower.

To determine the expression change of *MsFcγRIα*, *Syk* and *Lyn* with host defense against bacterial infection, LPS (50 ng/μL) and *N. seriolae* (1 × 10^8^ CFU/mL) were used to challenge the largemouth bass. And the identification of *N. seriola* was shown in Appendix A. In the spleen, after LPS stimulation, the expression of *MsFcγRIα* increased gradually and reached the peak value at 72 h p.i. (*p* < 0.01); *Syk* and *Lyn* expression levels reached the maximum at 48 h p.i. (*p* < 0.01), which was earlier than *MsFcγRIα.* In the head kidney, the highest mRNA expression of *Syk* was noticed at 6 h p.i. (*p* < 0.01) with *N. seriolae* stimulation; the mRNA expression of *MsFcγRIα* and *Lyn* reached the highest expression level at 24 h p.i. (*p* < 0.01) with *N. seriolae* or LPS stimulation. The peak value of gene expression with LPS stimulation was higher than with *N. seriolae* stimulation in the spleen, while the opposite is true in the head kidney (Figure 7B).

### 2.6. Variation of Genes in Leukocytes Post-Infection with LPS and N. seriolae

To compare the response pattern of *MsFcγRIα*, *Syk*, and *Lyn* under different immunogenic stimuli, leukocytes extracted from the head kidney were infected with LPS and *N. seriolae*, respectively. Peak induction levels of *MsFcγRIα*, *Syk*, and *Lyn* were observed at 72, 48, and 72 h p.i. with LPS stimulation (Figure 8A) and at 72, 48, 48 h p.i with *N. seriolae* stimulation (Figure 8A), respectively. Post *N. seriolae* infection at 48 h, the expression level of *Lyn* and *Syk* began to drop; Oppositely, at the same time, the expression of *MsFcγRIα* rapidly increased and at 72 h p.i and was 1.73-fold higher than *Lyn*. Under the stimulation of LPS or *N. seriolae*, the peak value of *Syk* expression was much earlier and higher than *MsFcγRIα.*

### 2.7. Enhancement of IMsFcγRIα on Reactive Oxygen Species and Phagocytosis in Leukocytes

To explore the effect I(r)MsFcγRIα on reactive oxygen species (ROS) and phagocytosis to *N. seriolae*, the leukocytes were incubated with 5 μg/mL recombinant MsFcγRIα-PET32a protein in advance and determined by flow cytometer. The results showed that RIin (r)MsFcγRIα-treated leukocytes was 33.87 ± 1.577%, which was increased significantly (*p* < 0.01) compared to the control group with 15.43 ± 0.12% (Figure 9A). Meanwhile, the phagocytic ability of leukocytes in the (r)MsFcγRIα-treated group with 60.27 ± 0.463% was significantly higher (*p* < 0.01) than the pET-32a (37.27 ± 0.480%) or PBS-treated group (37.13 ± 0.318%) (Figure 9B). It sugIted that (r)MsFcγRIα could promote the respiratory burst and phagocytosis of leukocytes.

### 2.8. TranscriptionalIdulation of (r)MsFcγRIα in Leukocytes

To further investigate the effect of MsFcγRIα on mRNA transcription of leukocytes, the relative expression levels of immune-related genes in leukocyteItimulated with (r)MsFcγRIα (5 μg/mL) were determined by qRT-PCR. The results (Figure 10) indicated that several of the genes related to phagocytosis, such as *Syk* (*p* < 0.01), *MsFcγRIα* (*p* < 0.01), *ATG3* (*p* < 0.01), *MARCKS* (*p* < 0.01), *Lyn* (*p* < 0.05) as well as genes engaged in the complement system, including *C1R* (*p* < 0.01), *C3* (*p* < 0.01), *C5* (*p* < 0.01) and *C7* (*p* < 0.01) were significantly up-regulated.

## 3. Discussion

FcRs carried out functions of interaction with immune complexes and hematopoietic cells have been extensively studied in mammalian leukocytes [28]. However, the immunoregulation of FcRs in teleost remains unclear. In this study, a largemouth bass FcR cDNA, termed MsFcγRIα, was identified and characterized. Diverse biological functions of FcγRs are mediated by its extracellular domains, which contain two or three Ig domains (i.e., D1, D2, and D3) [29], and the homology of the D1 domain and D2 domain in teleost and mammals are relatively higher than of that D3 domain [2,19,29]. MsFcγRIα, akin to PaFcγRI of ayu [19] and JF-FcR-like protein 1 of Japanese flounder [30], also possessed only two Ig domains with D1 and D2 clustered with the corresponding domain of other fish and mammalian FcγRs. Phylogenetic and blast analysis showed that MsFcγRIα shares a close homology to a group of teleost FcRs, such as Asian seabass (54.58% identity), large yellow croaker (49.60% identity), and other fish, indicating that FcRs of different teleost presumably exert similar functions. The study of MsFcγRIα may provide an important reference value for further research on FcRs of teleost.

Early studies have unveiled that most mammalian FcRs are membrane-bound proteins with TM domain or glycophosphatidyl inositol (GPI) [31,32,33]. Whereas, differently from mammalian FcRs, MsFcγRIα with a signal peptide at N-terminus lacks GPI-anchor or TM/CYT segments, predicting that MsFcγRIα was secreted and/or intracellularly expressed. The soluble FcRs absenting TM/CYT are generated by alternative splicing or proteolytic cleavage of the membrane FcRs and have been found in ayu [19], Japanese flounder [30], channel catfish [2,34], human [6,35] and mouse [36]. In this study, MsFcγRIα was mainly localized on the cytoplasm of HEK293 cells after transfection and also detectable in the supernatants of HEK293 cells and serum of largemouth bass infected with *N. seriolae.* Moreover, the results of IHC showed that native MsFcγRIα was remarkably distributed among the extracellular matrix in the largemouth bass spleen, head kidney, and intestine. A line of results collectively pointed to MsFcγRIα as a soluble protein that could be secreted through its signal peptide. Both membrane and soluble FcγRIα (CD64) have been found in humans [37]. However, the presence of membrane FcγRIα in largemouth bass still needs to be identified.

In previous studies, soluble FcRs have participated in mammalian B cell proliferation and antibody production [21]. Their role in immune response needs further study. The interaction of IgG and FcγR is the first step for particle internalization [38] and immunomodulatory mediated by the binding of IgG/IgM to soluble FcRs. This has been identified in the mouse [39,40], humans [37], ayu [19], and channel catfish [2,34]. The combination of IgM with MsFcγRIα was characterized in this study. The binding of IgM to MsFcγRIα was determined by a eukaryotic pDisplay system where transfected HEK293 cells expressed surface-targeted MsFcγRIα. A similar method using the eukaryotic expression vector to determine the combination of IgM with soluble FcRs has been reported in previous studies [2,41]. The results further illustrated [19,42] the identity between soluble FcRs and the corresponding membrane receptors in antigen recognition [19]. Moreover, the high affinity of IgM with soluble MsFcγRIα may be attributed to the D1 and D2 Ig domains in MsFcγRIα instead of N-linked glycosylation of MsFcγRIα [2].

*MsFcγRIα* is expressed predominantly in largemouth bass lymphoid tissues, including the spleen, head, kidney, and intestine. It is also detected in nonhemopoietic tissues such as the brain, liver, gill, and skin with relatively low expression levels, which was coincident with the expression of *IpFcRI* in channel catfish [2]. The spleen and head kidney are hematopoietic and immune defense organs in teleosts [43,44] and play an essential role in cytokine production [45]. Interestingly, in this study, the mRNA expression levels of all detected genes in the head kidney (*MsFcγRIα*, *Syk*, and *Lyn*) after *N. seriolae* infection were higher than in LPS infection within 72 h. Still, the tendency in the spleen showed otherwise, suggesting that the head kidney may be more sensitive to *N. seriolae* infections than the spleen in the acute phase response. Therefore, in the current study, as in many previous studies [46,47,48], leukocytes isolated from the head kidney were used for the investigation of the immune response. Upon LPS and *N. seriolae* infections, the expression level of *Syk* and *Lyn* began to increase at 24 h p.i. However, the beginning time of *MsFcγRIα* was 48 h p.i, later than *Syk* and *Lyn.* The gradual increase of *MsFcγRIα* may attribute to the activation of the FcγR-mediated phagocytosis pathway after crosslinking of ITAM in MsFcγRIα [49,50]. Once the FcRs bind to the antigen, tyrosine phosphorylation of Syk and Lyn occurs. The level of the phosphorylation then decreases during internalization [7]. Likewise, the expression level of *Syk* in this study declined at 48–72 h p.i. Furthermore, the expression level of *Syk* was distinctly higher than *Lyn* and *MsFcγRIα*, indicating that *Syk* and *Lyn* have different functions in the transduction of phagocytic signal generated by *MsFcγRIα* [7], and *Syk* is necessary for phagocytosis of leukocytes [51].

It has been elaborately investigated that the endocytosis of FcRs is dependent on clathrin and dynamin. Conversely, the cytoplasmic domain of FcRs in phagocytosis is unessential [52]. Therefore, whether the MsFcγRIα without TM/CYT affects phagocytosis remains a matter of debate. Generally, reactive oxygen species (ROS) are generated by phagocytes during the internalization process [53], and the level of ROS usually accounts for the bactericidal effect of leukocytes [54]. In this study, the flow cytometric assay showed that the ROS and phagocytic ability increased significantly with (r)MsFcγRIα treatment. Coincidentally, the mRNA expression of phagocytosis-related genes, including *Syk*, *Lyn*, *MsFcγRIα*, *ATG3*, and *MARCKS* was up-regulated remarkably, indicating that soluble MsFcγRIα could facilitate the endocytosis of leukocytes. This up-regulation is the opposite of the recognized inhibitory effect of soluble FcRs through competing with membrane FcRs for Ig binding [21,34]. However, on the one hand, the interaction of soluble FcγRIII with complement receptors (CR3 and CR4) can trigger cellular activation by promoting IL-6 and IL-8 production and regulating inflammatory processes [22]. On the other hand, the combination of FcγRIIA with CR3 also can enhance phagocytosis [55]. Thus, we hypothesized that the phagocytosis of leukocytes was mediated by the interaction of soluble MsFcγRIα with complement receptors, which was verified preliminarily by the significant up-regulation of mRNA expression of the complement system (*C1R*, *C3*, *C5*, and *C7*) with (r)MsFcγRIα stimulation in this study. However, the scarcity of information on the complement system in largemouth bass has seriously hindered the study of the phagocytosis mechanism in teleosts.

In summary, *MsFcγRIα* successfully cloned in this study was structurally conserved with FcRs homologs in fish and mammals and highly expressed in lymphoid tissues of largemouth bass. Soluble form MsFcγRIα was detected in *N. seriolae*-infected largemouth bass serum and the supernatants of transfected HEK293 cells, as well as the extracellular matrix of the intestine, spleen, and head kidney. MsFcγRIα could bind to IgM and play a crucial role in the early stage of immune response to bacteria. In addition, MsFcγRIα promoted the ROS activation and phagocytosis of leukocytes which may be related to the interaction of MsFcγRI with the complement receptor. This study lays a foundation for investigating soluble FcRs in ADCP and will be conducive to exploring the phagocytosis mechanism of leukocytes in teleost.

## 4. Materials and Methods

### 4.1. Microorganisms, Fish, and Cell Line

*N*. *seriolae* isolated from largemouth bass was identified by Gram’s stain, PCR, and DNA sequencing, cultured in Brain Heart Infusion (BHI) broth for four days at 25 °C, and enumerated before experiments. The healthy largemouth bass (15 ± 5 g) were purchased from a fish farm at Foshan, Guangdong province, China, and fed twice daily with commercial feed. The fish were randomly divided into a density of 45 fish per tank and acclimated in an automatic filtering aquaculture system at 26 ± 1 °C for 21 days. Prior to any animal experiment, the fish were anesthetized with 20 mg / L of eugenol. HEK 293 cells were collected from the laboratory of Dr. Li [56] and cultured in Dulbecco’s modified eagle medium (DMEM) supplemented with 100 U/mL penicillin, 100 μg/mL streptomycin and 10% fetal bovine serum (FBS) (Thermo Fisher Scientific, MA, USA) at 37 °C.

### 4.2. Cloning and Bioinformatics Analysis

The open reading frame (ORF) of *MsFcγRIα* assembled from a largemouth bass transcriptome data (NCBI Accession No.: PRJNA806622) was cloned. Primers, MsFcγRIα-OF and MsFcγRIα-OR (Appendix A) were designed by Primer Premier 5.0. The PCR products were ligated into pClone007 versatile simple vector (TSINGKE, China), then transformed into component *Escherichia coli* (*E. coli*) DH5α. Finally, three positive clones were sequenced by TSINGKE Biotechnology company (Guangzhou, China).

Bioinformatics analysis of MsFcγRIα was performed as the following methods. DNA sequences were analyzed by SnapGene Viewer (http://www.snapgene.com/ (accessed on 13 March 2019)). Multiple alignments of amino acid sequences were performed with DNAMAN 8.0 software (Lynnon Biosoft, QC, Canada). The similarity of the amino acid sequences was carried out by NCBI protein BLAST programs (http://blast.ncbi.nlm.nih.gov/Blast.cgi (accessed on 14 March 2019)). Phylogenetic trees were constructed by MEGA 6.0 software using the neighbor-joining (NJ) method with 1000 bootstrap replicates. The domain architecture was predicted by SMART program (http://smart.embl-heidelberg.de/ (accessed on 20 April 2019)). N-glycosylation sites were predicted by the NetNGlyc 1.0 server (http://www.cbs.dtu.dk/services/NetNGlyc/, SignaIP-5.0 (accessed on 6 August 2019)), and the signal peptide was analyzed by SignaIP (https://services.healthtech.dtu.dk/service.php?SignalP-5.0 (accessed on 12 January 2021)). Conserved cysteine residues were predicted by ExPASy-PROSITE (https://prosite.expasy.org/ (accessed on 22 January 2021)). Protein tertiary structure was predicted online via the ExPASy website (https://swissmodel.expasy.org/ (accessed on 25 January 2021)). Transmembrane domain prediction was conducted using TMHMM-2.0 (https://services.healthtech.dtu.dk/service.php?TMHMM-2.0 (accessed on 25 January 2021)). GPI-anchorage site was predicted by the big-PI Predictor (https://mendel.imp.ac.at/gpi/gpi_server.html (accessed on 28 January 2021)).

### 4.3. Transcriptional Expression Profile of Genes

To study the mRNA expression of *MsFcγRIα* in healthy largemouth bass, tissues including the spleen, head kidney, brain, intestine, gill, skin, muscle, and liver were collected from three individual fish. Additionally, Gene (*MsFcγRIα*, *Lyn*, *Syk*) expression levels of largemouth bass intraperitoneally injected with 100 μL LPS (50 ng/μL) (Sigma, Aldrich, USA), 100 μL *N. seriolae* (1 × 10^8^ CFU/mL), and 100 μL PBS (as the control group), respectively, were determined by quantitative real-time PCR (qRT-PCR). The spleen and head kidney tissues were isolated from three individual fish at 0, 3, 6, 12, 24, 48, and 72 h post-injection (h p i), then snap-frozen in liquid nitrogen and stored at −80 °C before RNA extraction.

Total RNA was extracted using E.Z.N.A.^®^ Total RNA Kit II (OMEGA, China), and cDNA was synthesized from 1 μg of total RNA with the HiScript^®^IIQ RT Supermix Kit (Vazyme, China). The 1:10 of diluted cDNA as templates for the gene expression analysis by qRT-PCR according to the manufacturer’s instructions (AceQ qPCR SYBR Green Master Mix, Vazyme). Primers designed using Primer Primier 5.0 are listed in Appendix A. The qRT-PCR program was 95 °C for 3 min, followed by 40 cycles of 95 °C for 15 s, 58 °C or 15 s, and 72 °C for 35 s and amplification products were produced and analyzed by CFX Connect Real-Time System (Bio-Rad, CA, USA). The fold changes of target genes were standardized against *β-actin* using the 2^−ΔΔCt^ method [57].

### 4.4. Leukocytes Isolation and Stimulation with LPS and N. seriolae

To detect the expression rules of *MsFcγRIα*, *Lyn*, and *Syk* under antigen stimulation, the leukocytes were isolated from the head kidney of largemouth bass by leukocytes isolation kit (TBD, China) as described in the manufacturer’s introduction. Briefly, the cell suspension from tissues was overlaid gently on the Ficoll for leukocytes, then centrifuged at 450× *g* for 25 min at 20 °C. Leukocytes were harvested from the intermediate layer and washed three times in RPMI-1640 (Gibco, CA, USA). Prior to resuspending the leukocytes in RPMI-1640 with 10% fetal bovine serum (FBS; Gibco, CA, USA) and 1% penicillin/streptomycin (Hyclone, Logan, UT, USA), the activity of cells was defined using 0.4% trypan blue (TBD, China). Then 1 mL of the cell suspension (1 × 10^7^ cells/mL) incubated with LPS (50 μg/mL), formalin-inactivated *N. seriolae* (1 × 10^8^ CFU/mL) and PBS (100 μL/mL) at 25 °C, respectively. Leukocytes were collected at 0, 3, 6, 12, 24, 36, 48, and 72 h post-injection (h p i) and centrifuged at 500× *g* for 10 min at 4 °C, then snap-frozen in liquid nitrogen and stored at −80 °C before RNA extraction.

### 4.5. Preparation of Recombinant Protein and Polyclonal Antibody (PcAb)

The CDS sequence encoding mature protein but excluding the signal peptide was amplified by expression primers (MsFcγRIα-32a-F and MsFcγRIα-32a-R) (Appendix A). The amplified product was purified and ligated to the BamH I and Hind III (TaKaRa, Japan) restriction enzymes of PET-32a vector (TaKaRa, Japan) using T4 DNA Ligase (TaKaRa, Japan). Then, the ligation product was transformed into *E. coli* TSsetta (DE3) (TSINGKE, China), induced by 1 mM isopropyl-β-D-thiogalactopyranoside (IPTG) with O.D. 600 of 0.6 and cultured at 26 °C for 12 h. The protein purification was performed according to the protocol of His-tag Protein Purification Kit (Beyotime, China) and dialyzed at 4 °C in urea with gradient concentrations, as well as PBS. Eventually, the purified recombinant MsFcγRIα ((r)MsFcγRIα) was concentrated by PEG2000 (Dingguo, China) and identified by SDS-PAGE gel electrophoresis.

To prepare polyclonal antibody (PcAb) against protein MsFcγRIα (anti-MsFcγRIα), healthy six-week-old female BALB/C mice were immunized with the purified recombination protein. The mice were intraperitoneally injected with 50 μg/100 μL MsFcγRIα per mouse each time at two-week intervals for four times. Except for the first-time immunization, the MsFcγRIα was emulsified with complete Freund’s adjuvant (Sigma, Aldrich, USA), and each subsequent immunization was carried out with incomplete Freund’s adjuvant (Sigma, Aldrich, USA) instead. Four days after the final immunization, blood was obtained from the retro-orbital of the mice following the isoflurane anesthesia, then collected the serum by centrifugation at 5000× *g*, 4 °C for 10 min, then purified by Melon Gel Monoclonal IgG Purification Kit (Thermo Fisher Scientific, MA, USA). Prior to being equally stored at −80 °C, the specificity was characterized by Western blotting.

### 4.6. Immunohistochemistry (IHC)

Immunohistochemistry (IHC) was performed as previously described with some modifications [58]. Briefly, sections of the intestine, spleen, and head kidney from largemouth bass were made by dehydration and paraffin-embedded treatment following 4% paraformaldehyde fixation. Then, the sections were subjected to antigen retrieval by microwave for 10 min at 100 °C in sodium citrate buffer (pH 6.0) and washed with PBS. Endogenous peroxidases were removed in 3% H_2_O_2_ and 85% methanol for 15 min. Following the blocking with 1% BSA for 1 h, the sections were overnight incubated with anti-MsFcγRIα PcAb at a dilution of 1:500 at 4 °C, and PBS was used as a negative control. After washing with PBS, slides were incubated with the horseradish peroxidase-conjugated (HRP) goat anti-mouse immunoglobulin G (IgG) secondary antibody (Sigma, Aldrich, USA) at a dilution of 1:2000 for 1 h at 37 °C. DAB detected the signal expression of MsFcγRIα under the microscope (Leica, Germany), then slides were counterstained in 50% Hematoxylin and washed with 0.02% ammonia water. After dehydration by 70% ethanol and 95% ethanol, slides were incubated in xylene for 5 min and sealed with neutral gum. PBS solution instead of anti-MsFcγRIα PcAb was used as a negative control. Images were obtained with a Leica DM2500 microscope (Leica, Germany).

### 4.7. Transfection and Location of MsFcγRIα

To locate the expression of MsFcγRIα, MsFcγRIα containing signal peptide was amplified using the primers, MsFcγRIα-EGFP-F and MsFcγRIα-EGFP-R (Appendix A). PCR products were recovered with a DNA gel extraction kit (Axygen, USA) and ligated into PEGFP-N1 with Hind III and BamH I (TaKaRa, Japan) restriction enzymes. Then, the recombinant eukaryotic plasmid (MsFcγRIα-EGFP) was extracted by the EndoFree plasmid Kit (Tiangen, China) to remove endotoxin and transiently transfected into HEK293 cells on a 15 mm confocal dish (Biosharp, China) (3 × 10^5^ cells/well) by Lipofectamine 3000 (Invitrogen, USA). After 24 h, transfected cells were stained with DiI by Cell Plasma Membrane Staining Kit (Red Fluorescence) (Beyotime, Shanghai, China) at 37 °C for 20 min and 10 μg/mL DAPI (Sigma, Aldrich, USA) for 10 min. Cells were observed under a laser scanning confocal microscope (TCS SP8 Leica, Germany).

Furthermore, to test whether MsFcγRIα could generate a secretory protein, the recombinant eukaryotic plasmid (MsFcγRIα-pcDNA3.1(+)) was constructed. Primers (MsFcγRIα-3.1-F and MsFcγRIα-3.1-R) used to amplify the full length of MsFcγRIα CDS were listed in Appendix A. The recombinant plasmid was transfected into HEK293 cells as described above. After 24 h, the expression of native MsFcγRIα was determined by anti-MsFcγRIα PcAb (1:500 *v*/*v*) and Alexa Fluor^®^ 647 goat anti-mouse IgG (Abcam, UK) (1:2000 *v*/*v*) with flow cytometry (BD FACS Calibur, San Jose, CA, USA), and laser scanning confocal microscope (TCS SP8 Leica, Germany). The supernatants and cell lysates were collected independently to determine secretory MsFcγRIα by Western blotting.

### 4.8. Binding of MsIgM to MsFcγRIα by FACS and IIFA

To establish whether the surface-expressed MsFcγRIα could bind to largemouth bass IgM (MsIgM), the IgM was purified from largemouth bass serum by HiTrap^®^ Protein A column (GE Healthcare, Lafayette, CO, USA) according to the previous procedure [59]. The interaction between MsFcγRIα and MsIgM was identified as described in previous studies with some modifications [19,42]. Plasmid (MsFcγRIα-EGFP) was transfected into HEK293 cells in 6-well plates as described above. Plasmid (pEGFP-N1) was equally transfected as a negative control. After 24 h, the complete medium was replaced with serum-free DMEM, then 20 μg/mL purified MsIgM was added for 0 h, 6 h, 12 h, or 24 h at 37 °C. The transfected cells (2 × 10^6^ cells/mL) were harvested and washed with PBS prior to incubation with anti-MsIgM mAb of largemouth bass (MM06H) (1:1000 *v*/*v*) (Wu et al., 2022) for 1 h at 37 °C. After washing with PBS, the cells were labeled with Alexa Fluor^®^ 647 goat anti-mouse IgG (Abcam, UK) for 1 h at 37 °C. Finally, cells were resuspended in PBS following the three times washing and analyzed by flow cytometry (BD FACS Calibur, San Jose, CA, USA). Meanwhile, 500 μL of transfected cells settled onto a 15 mm glass confocal dish (Biosharp Life Science, Korea) were treated with 20 μg/mL MsIgM and labeled with Alexa Fluor^®^ 647 goat anti-mouse IgG (Abcam, UK) as mentioned above. After staining with 10 μg/mL DAPI (Sigma, Aldrich, USA), fluorescence images were acquired using a laser scanning confocal microscope (TCS SP8 Leica, Germany).

### 4.9. Co-Immunoprecipitation and Western Blotting

Co-immunoprecipitation experiment was performed to confirm the interaction between the native MsFcγRIα with MsIgM according to the previous study with some modifications [15]. Briefly, HEK 293 cells in 6-well plates (2 × 10^6^ cells/well) were transfected with 2.5 μg MsFcγRIα-pcDNA3.1(+) per well and treated with 20 μg/mL MsIgM for 24 h as described above. Following careful washing with pre-cooling PBS, the cells were lysed with 200 μL immunoprecipitation lysis buffer (Beyotime, China) with 1 mm PMSF (Dingguo, China) on ice. Cellular debris was removed by centrifugation at 12,000× *g* for 5 min at 4 °C. The cell supernatants were incubated overnight with MM06H mAb (1:1000) at 4 °C. Non-immune mouse serum instead of MM06H mAb was performed as a negative control. 40 μL protein A+G agarose (Beyotime, China) was added into the mixture for incubation with slow rotation at 4 °C for 3 h. After five times washing with PBS by centrifugation at 1000× *g* for 5 min at 4 °C, the samples were boiled with 40 μL 1× SDS loading buffer for 5 min and used for immunoblot analysis with MM06H mAb and anti-MsFcγRIα PcAb.

For Western blotting analysis, protein samples were separated using 12% SDS-PAGE gels and transferred onto a 0.22 μm pore size polyvinylidene difluoride (PVDF) membranes (Millipore, MA, USA) by 160 mA for 90 min. The PVDF membranes were blocked in 5% (*w*/*v*) skim milk in TBST (10 mM Tris-HCl, 150 mM NaCl, 0.05% Tween-20; pH 7.4) for 1 h at 37 °C, then incubated with MM06H mAb (1:1000) or anti-MsFcγRIα PcAb (1:500 *v*/*v*) for overnight at 4 °C in TBST containing 0.5% (*w*/*v*) skim milk. Then, the membranes were washed with TBST three times and incubated with HRP-conjugated goat anti-mouse IgG (1:5000) (Sigma, Aldrich, USA) in TBST containing 0.5% (*w*/*v*) skim milk for 1 h at 37 °C. After washing three times with TBST, the membranes were stained with ECL detection reagents (Thermo Fisher Scientific, MA, USA). Staining was stopped with distilled water, and blots were analyzed with a gel imager (Tanon, China).

### 4.10. Detection of Reactive Oxygen Species (ROS)

According to the manufacturer’s instructions, reactive oxygen species (ROS) were detected by a ROS assay kit (Beyotime Biotech, China). Briefly, the leukocytes isolated from largemouth bass were resuspended and adjusted to 1 × 10^6^ cells/mL with RPMI1640 culture medium (Gibco, CA, USA) and then stimulated with 5 μg/mL recombinant MsFcγRIα-PET32a ((r)MsFcγRIα) or equal PET32a protein or PBS in 24-well plates containing 1 mL culture medium per well at 25 °C for 3 h. Following three times washing with PBS, the cells were harvested and stained with 10 μM DCFH-DA at 37 °C for 20 min in the dark. The fluorescence intensity was determined by flow cytometry (BD FACS Calibur, San Jose, CA, USA). Each experiment was conducted in triplicates for each independent sample.

### 4.11. Phagocytosis Assay

Phagocytosis assay was performed as described in the previous study with some modifications [19,60]. Briefly, the leukocytes from largemouth bass were plated in 24-well plates at a cell density of 5 × 10^6^ cells/well with RPMI1640 culture medium (Gibco, USA) and then incubated with 5 μg/mL (r)MsFcγRIα or equal PET32a or PBS at 25 °C for 3 h. The *N. seriolae* was inactivated by 0.2% formaldehyde at 4 °C overnight, then labeled with 0.2 mg/mL FITC (Thermo Scientific, MA, USA) at 4 °C for 12 h. The FITC-labeled *N. seriolae* was added to cells at a cell-to-bacteria ratio of 1:20 for incubation at 20 °C or without bacteria treatment as a negative control. After 6 h, the non-ingested bacteria were removed by ficoll kit, and cells were harvested by centrifugation at 450× *g* for 20 min at 20 °C. Following three washing steps with PBS, cells were resuspended in PBS. Flow cytometry (BD FACS Calibur, San Jose, CA, USA) analyzed the phagocytic ability of leukocytes. The relative gene expression of leukocytes after (r)MsFcγRIα stimulation was determined by qRT-PCR as described in Section 4.3. Each experiment was conducted in triplicates for each independent sample.

### 4.12. Statistical Analysis

Statistical analysis was conducted by Statistical Product and Service Solution (SPSS) software (version 20.0; IBM Corp., USA), GraphPad Prism software (version 8.0.1; San Diego, CA, USA), and Flowjo V10 software (version 10; Ashland, USA). The statistical significance was analyzed using one-way ANOVA and considered no significant at *p* > 0.05, * significant at *p* < 0.05, and **** significant at *p* < 0.01. The results of gene relative expression were presented as mean ± standard deviation (SD), and the percentage of cells was presented as mean ± SEM.

## Figures and Tables

**Figure 1 ijms-23-13788-f001:**
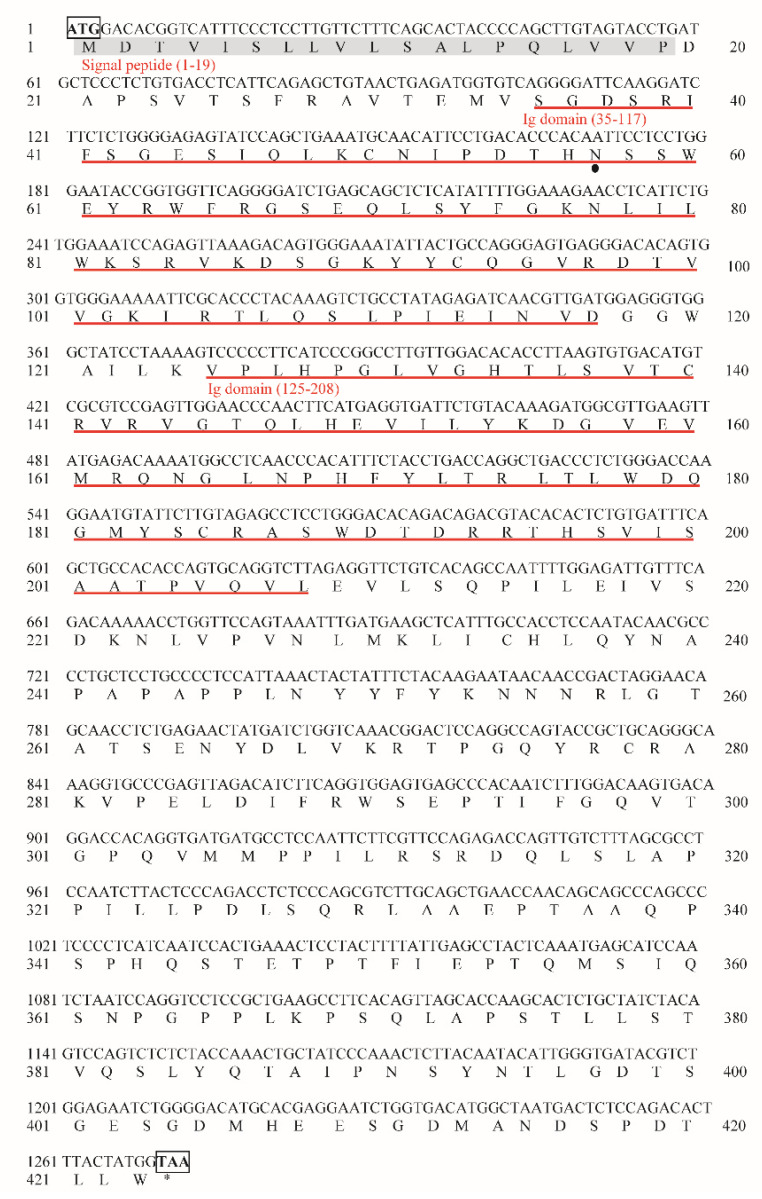
Nucleotide and deduced amino acid sequence of *MsFcγRIα* (GenBank accession No. OK258092). The predicted signal peptide domain is shaded in gray, and two Ig domains are indicated with red underlines. The solid black dot highlights a potential N-linked glycosylation site. The start codon (ATG) and stop codon (TAA) are marked with black boxes, and the stop codon was also illustrated as “*”.

**Figure 2 ijms-23-13788-f002:**
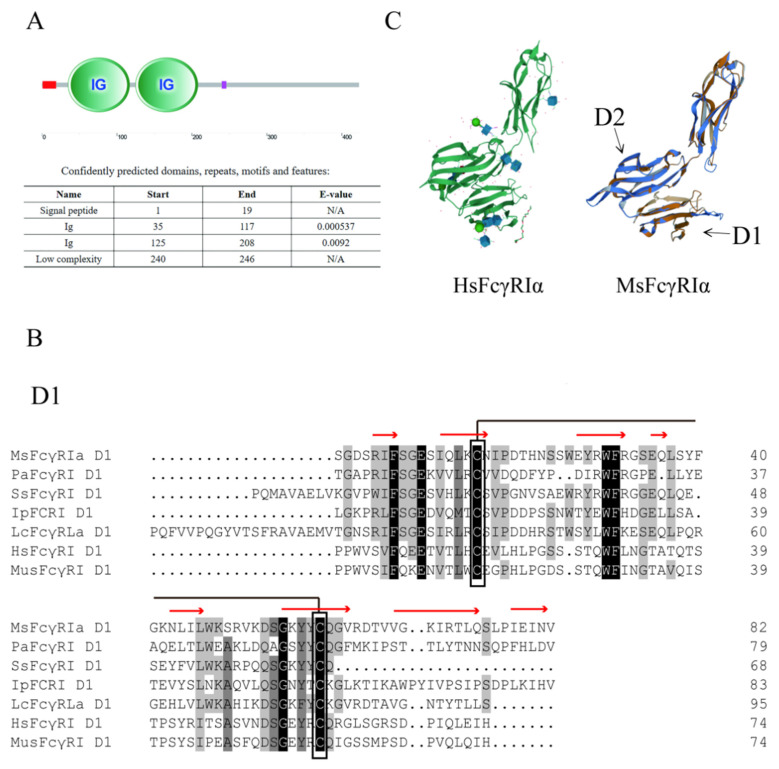
Analysis of MsFcγRIα protein sequences. (**A**) Schematic representation of MsFcγRIα protein. The red line on the left and the purple line on the right represents signal peptide and low complexity domain, respectively. (**B**) Amino acid alignments of D1 and D2 of largemouth bass, ayu, Atlantic salmon, channel catfish, large yellow croaker, human and mouse representative FcγRs. Hatched boxes indicate conserved cysteines, and dashes (-) represent gaps. MsFcγRIα: Largemouth bass FcγRIα (ULE36159.1); PaFcRI: Ayu (MG687271); SsFcγRI: Atlantic salmon FcγRI (XM014162558); IpFcRI: Channel catfish FcRI (DQ286290); LcFcRLα: Large yellow croaker FcRLa (XM010738182); HsFcγRI: Human FcγRI (L03418); MusFcγRI: Mouse FcγRI (AF143170). Red arrows represent the predicted *β*-strand for MsFcγRIα. (**C**) Structural comparison of MsFcγRIα with HsFcγRIα (PDB; 3RJD, left panel). The aligned residues are shown in orange and blue, while the parts that are not aligned are shown in lighter shades of the same colors.

**Figure 3 ijms-23-13788-f003:**
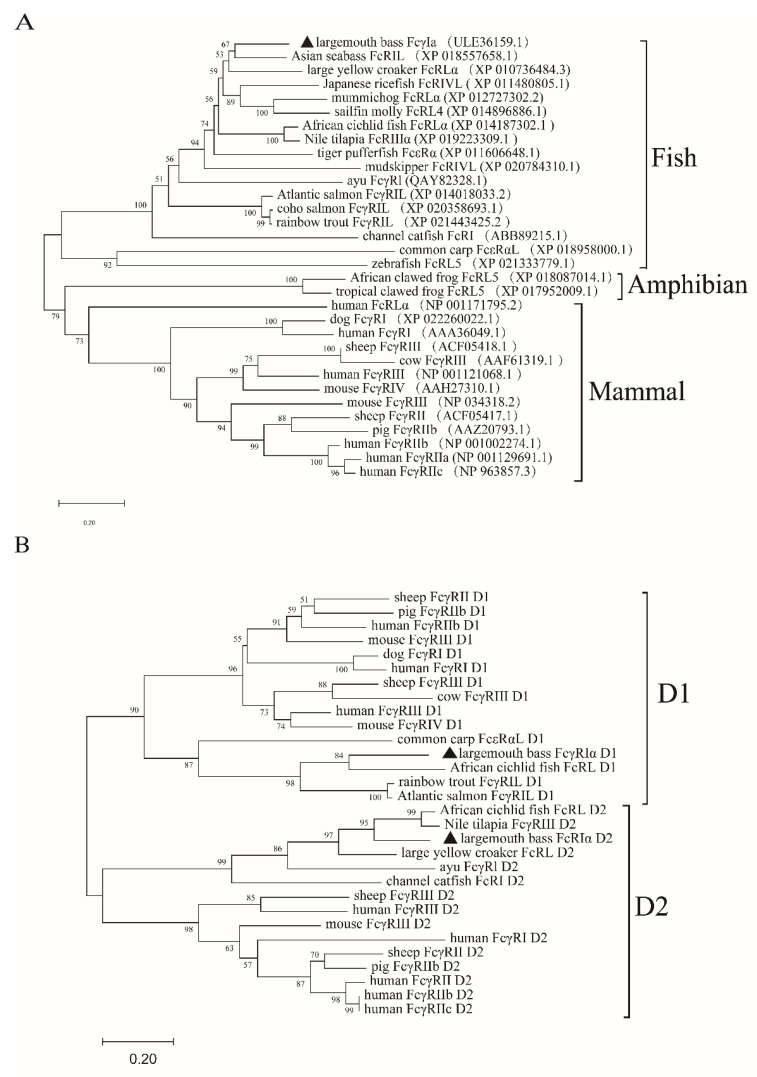
Phylogenetic tree of FcRs of different species. Full-length (**A**) and two Ig domains (**B**) Amino acid sequences of MsFcγRIα and other vertebrate FcRs were analyzed by the Neighbor-joining method based on the alignment with the Clustal W method. D1 and D2 represent the Ig domain 1 and domain 2, respectively. Numbers at forks show the percentage of branches in which this grouping occurred after bootstrapping on 1000 replicates using the MEGA X program. The scale bar indicates the number of substitutions per base.

**Figure 4 ijms-23-13788-f004:**
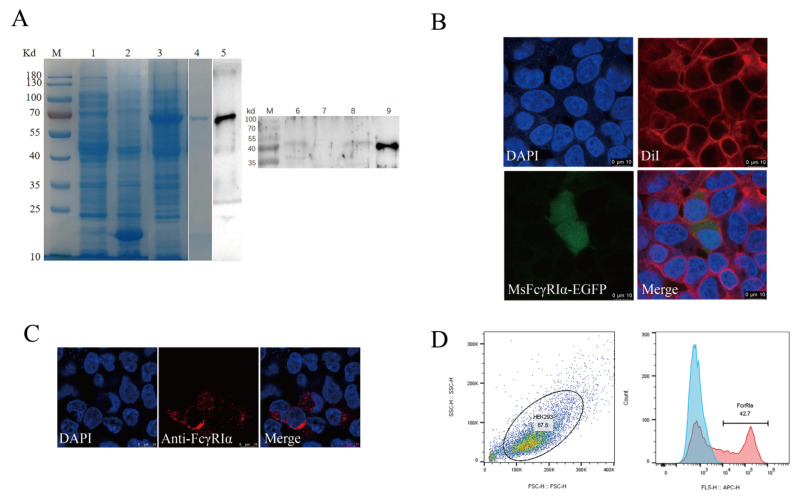
Antibody specificity and subcellular localization of MsFcγRIα. (**A**) SDS-PAGE and Western blotting analysis of MsFcγRIα. Lane M, molecular mass marker; lane 1, transformed *E. coli* without IPTG induction; lane 2, PET32a vector induced with IPTG; lane 3, MsFcγRIα-PET32a protein induced with IPTG; lane 4, SDS-PAGE of purified MsFcγRIα-PET32a protein (64.6 kDa); lane 5, Western blotting of MsFcγRIα-PET32a protein (64.6 kDa); line 6, Native MsFcγRIα in largemouth bass serum with *N. seriolae* immunization (47 kDa); line 7, Native MsFcγRIα in largemouth bass serum without *N. seriolae* immunization; lane 8, Supernatants of HEK293 cells transfected with FcγRIα-pcDNA3.1(+) (47 kDa); line 9, Cell lysates of HEK293 cells transfected with MsFcγRIα-pcDNA3.1(+) (47 kDa). (**B**) Subcellular localization of MsFcγRIα on HEK293 cells; Green fluorescence representing MsFcγRIα-EGFP is mainly located on cytoplasm and cell membrane. Scale bar = 10 μm. (**C**) Detection of MsFcγRIα-pcDNA3.1(+) on transfected HEK293 cells by anti-MsFcγRIα PcAb. The red signal means that MsFcγRIα is successfully expressed and specifically recognized by anti-MsFcγRIα PcAb. Scale bar = 10 μm. (**D**) Flow cytometric analysis of the percentage of HEK293 cells expressing MsFcγRIα after 24 h transfection.

**Figure 5 ijms-23-13788-f005:**
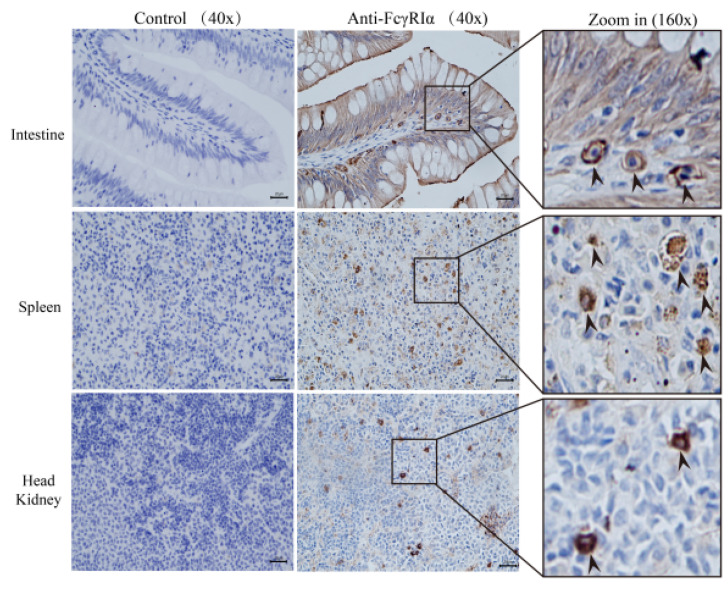
Secretory MsFcγRIα expression in largemouth bass tissues by IHC. The brown sediment suggests the presence of MsFcγRIα. Strongly positive signals (+++) are shown with arrows in the intestine, spleen, and head kidney, and no signal in the negative control. Positive signals are found among the cytoplasm in these three tissues. IHC results indicate that MsFcγRIα is secretory. IHC results are shown at 40× and 160× magnification. Scale bar = 20 μm.

**Figure 6 ijms-23-13788-f006:**
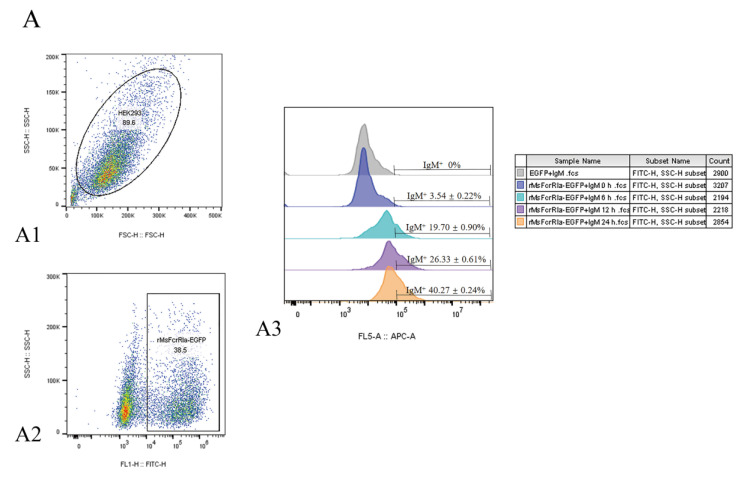
The interaction of MsIgM with MsFcγRIα. (**A**) Flow cytometry assay. (**A1**): FSC/SSC plots HEK293 cells. (**A2**): Gate of the HEK293 cells expressing green fluorescence after 24 h transfection represents the efficiency of transfection. (**A3**): Percentage of red fluorescence on pEGFP-N1-transfected (negative control) or MsFcγRIα-transfected HEK293 cells with 20 μg/mL MsIgM incubation for 0 h, 6 h, 12 h, 24 h, respectively, *n* = 3 (**B**) Co-localization of MsFcγRIα with MsIgM in HEK293 cells. The pEGFP-N1-transfected (negative control) or MsFcγRIα-EGFP expressed HEK293 cells were incubated with 20 μg/mL MsIgM for 24 h and detected by anti-MsIgM primary mAb (MM06H) and second antibody Alexa Fluor^®^ 647 goat anti-mouse IgG. DAPI staining shows the location of the nucleus. Scale bar = 10 mm. (**C**) Co-immunoprecipitation assay. The pcDNA3.1(+)-transfected (negative control) or MsFcγRIα-pcDNA3.1(+) transfected HEK293 cells were incubated with 20 μg/mL MsIgM for 24 h (without MsIgM incubation as a negative control). Protein extracts were incubated with MM06H, followed by Western blotting using MM06H and anti-MsFcγRIα.

**Figure 7 ijms-23-13788-f007:**
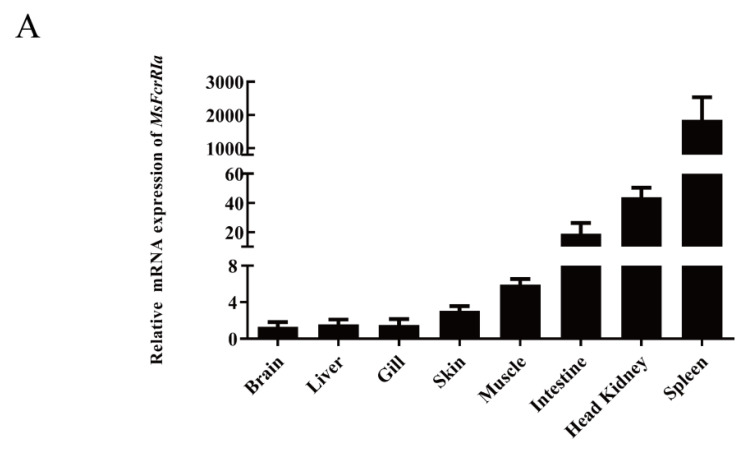
Relative expression analysis of genes in tissues. (**A**) Expression of *MsFcγRIα* in healthy largemouth bass tissues. Gene expression levels were normalized by *β-actin* and compared with the expression level in the brain, *n* = 3. (**B**) The mRNA level of *MsFcγRIα*, *Syk*, and *Lyn* after LPS (50 ng/μL) or *N. seriolae* (1 × 10^8^ CFU/mL) stimulation in the largemouth bass spleen and head kidney. Gene expression levels were normalized by *β-actin.* Data are expressed as the mean ± SD compared with the control group (PBS-stimulated), and asterisks show a significant difference (** p* < 0.05, *** p* < 0.01). The data were averaged from three independent experiments with three parallel repeats. *n* = 3.

**Figure 8 ijms-23-13788-f008:**
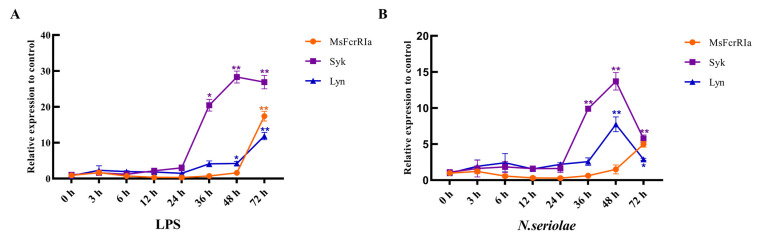
Relative expression analysis of genes in leukocytes with LPS (50 ng/μL) (**A**) or *N. seriolae* (1 × 10^8^ CFU/mL) (**B**) stimulation. Gene expression levels were normalized by *β-actin.* Data are expressed as the mean ± SD compared with the control group (PBS-stimulated), and asterisks show a significant difference (** p* < 0.05, *** p* < 0.01). The data were averaged from three independent experiments with three parallel repeats, *n* = 3.

**Figure 9 ijms-23-13788-f009:**
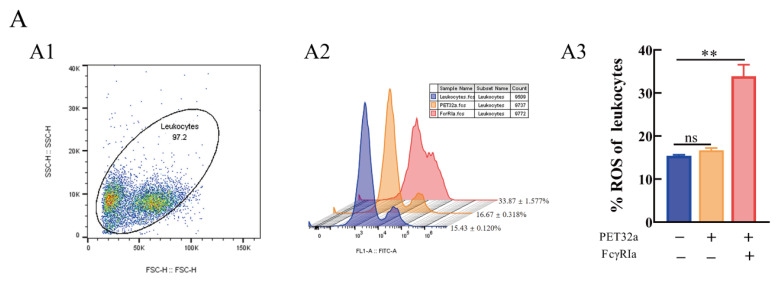
Effects of (r)MsFcγRIα on ROS (**A**) and phagocytosis to *N. seriolae* (**B**) of largemouth bass leukocytes by flow cytometric. (**A1**/**B1**): FSC/SSC plots leukocytes; (**A2**/**B2**): The histogram of flow cytometric analysis of the leukocytes ROS/phagocytosis pre-incubated with PBS, pET-32a protein or recombinant MsFcγRIα (5 μg/mL). The results shown here were one of three independent experiments; (**A3**/**B3**): The histogram of the ROS/phagocytosis rates. Three biological replicates were performed, and the data are presented as mean ± SEM (*n* = 3). *p* > 0.05, not significant (ns); significant; *p* < 0.01 (**), extremely significant.

**Figure 10 ijms-23-13788-f010:**
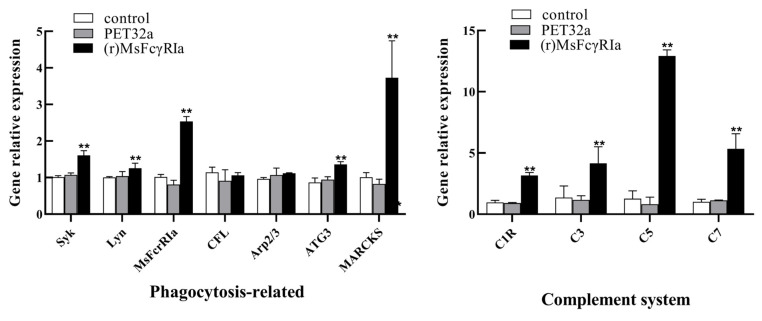
Analysis of gene expression in the leukocIs stimulated with (r)MsFcγRIα. The relative mRNA expression level of several genes related to phagocytosis (*Syk*, *Lyn*, *FcγRIα*, *CFL*, *Arp2/3*, *ATG3*, *MARCKS*) and complement system (*C1R*, *C3*, *C5*, *C7*) in leukocytes after recombinant MsFcγRIα (5 μg/mL) incubation were determined by qRT-PCR. Three biological replicates were performed, and the data are presented as mean ± SD (*n* = 3). *p* > 0.05, not significant; *p* < 0.05; *p* < 0.01 (**), extremely significant.

## Data Availability

Not applicable.

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
