# Peer review of "Functional Characterization of Largemouth Bass (Micropterus salmoides) Soluble FcγR Homolog in Response to Bacterial Infection"

_ijms, 2022, doi:10.3390/ijms232213788_

Round 1

Reviewer 1 Report

Jing Wu et al. Cloned the soluble form of FcγRIα receptor from largemouth bass. Expressed the FcγR protein in bacterai and prepoared a polyclonal antibody, evaluated the subcellular and tissular expressions of this soluble FcγRIα and investigated fucntioanl aspect of this ptotein.

The design of the experiments is good and appropriate, data sound and support the drawn coclusions.

Fcγ receptor I (FcγRI or CD64) is a high affinity for monovalent IgG and is expressed on immune cells. Activation of this receptor is finely regulated to avoid immune responses by non-antigen bound antibodies. However, the role of FcγRI cytoplasmic domain is not really understood. FcγRI binding to the monomeric IgG is necessary for the full interferon γ receptor signaling and its interaction with a larger high-avidity complexes can result in phagocytosis.

In this paper, the authors cloned the largemouth bass FcγRIα (MsFcγRIα) gene, expressed its recombinant protein in E. coli and prepared an antibody against FcγRIα. The authors also investigated the regulation of MsFcγRIα on respiratory burst, phagocytosis and transcriptional level of phagocytosis-related complement genes. Experiments are well conducted and the data are original for this teleost model with a good scientific background.

Author Response

Dear reviewer, 

Thank you for your hard work in reviewing our manuscript. It is our great honor to receive your approval!

In order to improve the quality of the article, we have made a minor modification in language with red font in our revised manuscript.

Wish these changes colud be accepted.

Sincerely,

Yugu Li

Reviewer 2 Report

Functional characterization of Largemouth bass (Micropterus salmoides) soluble FcγR homolog in response to bacterial infection. Overall well-written manuscript can be accepted with minor revision. 

Author Response

Dear reviewer, 

Thank you for your recognition of our work. To improve  the quality of our manuscript, we have made minor changes in writing and replace a high quality figure in Figure 4, please review this change in revised mansucript. We hope these changes colud be accepted.

Best wishes,

Yugu Li

Reviewer 3 Report

Functional characterization of Largemouth bass (Micropterus 1 salmoides) soluble FcγR homolog in response to bacterial infec-2 tion

Show the detection histograms of leukocyte populations, when the cells are extracted with Ficoll by centrifugation, the cell layer contains lymphocytes and monocytes, the population of neutrophils sediments together with the erythrocytes.

In phagocytosis experiments, the only population that phagocytoses is that of monocytes, therefore the value that is observed is a value of monocytes. This problem can be solved in several ways

1 Lying peripheral blood with commercial erythrocyte lysing solution, in that case FSC vs SSC morphology will clearly separate lymphocytes, monocytes and neutrophils.

2 Separating the population with Ficoll but adding a pan-leukocyte antibody such as CD45 (or the equivalent in their study) or a monocyte-specific antibody.

In Figure 4C, the quality of the photo has to be improved since it is not observed that there are 42.78% of positive cells as shown in Figure 4D.

In Figure 9 A1 and B1 of morphology, it is not possible to distinguish which population is lymphocytes and which population is monocytes.

If instead of working with mononuclear cells isolated with Ficoll we work with lysed blood only erythrocytes and keep the leukocytes alive, we will clearly have the three leukocyte populations, the neutrophils are recovered, which is important in the case of phagocytosis and oxidative burst.

 In line 288 replace a flow cytometry with by flow cytometry.

Round 2

Reviewer 3 Report

the answer is attached in the pdf

Author Response

Dear reviewer:

Thank you very much for your hard work in reviewing this article. we are sorry, we don’t know why there were no reference to the comments has been included in the first revision, and we will response them here.

Your suggestions are very valuable for us to isolate and clearly distinguish the leukocytes. We completely agree with your perception that the isolated cells only contain lymphocytes and monocytes, the neutrophils sediments will together with the erythrocytes, when the cells are extracted with Ficoll.

Working with peripheral blood using a buffer that lyses erythrocytes is a good suggestion, and we have tried it in our previous work. But as the different osmotic pressure of largemouth bass leukocytes from mammals and other fish, and there were little reference has been reported. So the PH and working time of lyses erythrocytes buffer is difficult control.

Regarding your second suggestion that separating the population with Ficoll but adding a pan-leukocyte antibody such as CD45 is a great idea. However, there may be difficult to handle with it due to the limit of pan-leukocyte antibody of largemouth bass.

And, in this research, we isolated the leukocytes from head kidney of largemouth bass by leukocytes isolation kit of fish tissues (WBC1080FZ, TBD, China), instead of Ficoll. And the result (Figure 9A1, B1) of our isolation was consistent with rainbow trout (Oncorhynchus mykiss)[1] (Red box shown in Figure 1 in attachment), which contains two populations including lymphoid gate (FSC-Hlow, SSC-Hlow) and myeloid cells (FSC-Hhigh; SSC-Hhigh). And the result of our separation of populations has shown in Figure 2 in attachment.

Thus, the method of isolation the leukocytes in this research can still support the argument of this article. In our further study, we will continue to improve our separation method as you suggested.

It is well recognized that "professional phagocytes", including macrophages/monocytes, neutrophils, and dendritic cells. Recent studies also have identified the potential for the phagocytosis of lymphacytes cells in reptiles[2], mice[3,4], humans[5] and teleosts[6]. And the purpose of this study is to investigate the effects of FcγR on phagocytosis of leukocytes. Thus, in Figure 9 A1 and B1, we gated the whole isolated leukocytes for analysis.

Reference:

  1. Korytář T, Jaros J,Verleih M, et al. 2013. Novel insights into the peritoneal inflammation of rainbow trout (Oncorhynchus mykiss). Fish Shellfish Immunol. 35 (4):1192-9. doi:10.1016/j.fsi.2013.07.032
  2. Zimmerman LM., Vogel LA., Edwards KA., et al., 2010. Phagocytic B cells in a reptile. Biol. Lett. 6, 270-273. https://doi.org/10.1098/rsbl.2009.0692
  3. Gao J., Ma X.,Gu W., et al., 2012. Novel functions of murine B1 cells: Active phagocytic and microbicidal abilities. Eur. J. Immunol. 42, 982-992. https://doi.org/ 10.1002/eji.201141519.
  4. Nakashima M., Kinoshita M., Nakashima H., et al., 2012. Pivotal advance: Characterization of mouse liverphagocytic B cells in innate immunity. J. Leukoc. Biol. 91, 537-546. https://doi.org/10.1189/jlb.0411214
  5. Zhu, Q., Zhang, M., Shi, M., Liu, Y., et al. Human B cells have anactive phagocytic capability and undergo immune activation upon phagocytosis of Mycobacterium tuberculosis. Immunobiology. 221, 558-567. https://doi.org/10.1016/j.imbio.2015.12.003
  6. Li J,Barreda DR, Zhang YA, et al. 2006. B lymphocytes from early vertebrates have potent phagocytic and microbicidal abilities. Nat Immunol. 7 (10):1116-24. doi:10.1038/ni1389
